# Exploratory Analysis of Sprint Force-Velocity Characteristics, Kinematics and Performance across a Periodized Training Year: A Case Study of Two National Level Sprint Athletes

**DOI:** 10.3390/ijerph192215404

**Published:** 2022-11-21

**Authors:** Dylan Shaun Hicks, Claire Drummond, Kym J. Williams, Roland van den Tillaar

**Affiliations:** 1SHAPE Research Centre, Flinders University, Bedford Park 5042, Australia; 2Department of Sport Science and Physical Education, Nord University, 7601 Levanger, Norway

**Keywords:** force, velocity, power, sprint, training, biomechanics, profile

## Abstract

**Objective:** This case study aimed to explore changes to sprint force-velocity characteristics across a periodized training year (45 weeks) and the influence on sprint kinematics and performance in national level 100-meter athletes. Force-velocity characteristics have been shown to differentiate between performance levels in sprint athletes, yet limited information exists describing how characteristics change across a season and impact sprint performance, therefore warranting further research. **Methods:** Two male national level 100-meter athletes (Athlete 1: 22 years, 1.83 m, 81.1 kg, 100 m time: 10.47 s; Athlete 2: 19 years, 1.82 cm, 75.3 kg, 100 m time: 10.81 s) completed 12 and 11 force-velocity assessments, respectively, using electronic timing gates. Sprint mechanical characteristics were derived from 30-meter maximal sprint efforts using split times (i.e., 0–10 m, 0–20 m, 0–30 m) whereas step kinematics were established from 100-meter competition performance using video analysis. **Results:** Between the preparation (PREP) and competition (COMP) phase, Athlete 1 showed significantly large within-athlete effects for relative maximal power (P_MAX_), theoretical maximal velocity (v_0_), maximum ratio of force (RF_MAX_), maximal velocity (V_MAX_), and split time from 0 to 20 m and 0 to 30 m (−1.70 ≤ ES ≥ 1.92, *p* ≤ 0.05). Athlete 2 reported significant differences with large effects for relative maximal force (F_0_) and RF_MAX_ only (ES: ≤ −1.46, *p* ≤ 0.04). In the PREP phase, both athletes reported almost perfect correlations between F_0_, P_MAX_ and 0–20 m (r = −0.99, *p* ≤ 0.01), however in the COMP phase, the relationships between mechanical characteristics and split times were more individual. Competition performance in the 100-meter sprint (10.64 ± 0.24 s) showed a greater reliance on step length (r ≥ −0.72, *p* ≤ 0.001) than step frequency to achieve faster performances. The minimal detectable change (%) across mechanical variables ranged from 1.3 to 10.0% while spatio-temporal variables were much lower, from 0.94 to 1.48%, with Athlete 1 showing a higher ‘true change’ in performance across the season compared to Athlete 2. **Conclusions:** The estimated sprint force-velocity data collected across a training year may provide insight to practitioners about the underpinning mechanical characteristics which affect sprint performance during specific phases of training, plus how a periodized training design may enhance sprint force-velocity characteristics and performance outcomes.

## 1. Introduction

Across a training year, sprint athletes typically progress through a periodized training program aimed at peaking towards major competitions including national championships. Training components within a sprint program generally include acceleration and maximal velocity sprinting, resistance training and plyometrics [1] which aim to enhance neuromuscular, biomechanical and technical sprint characteristics. However, the overall aim of all sprint programs should be to improve an athlete’s ability to run fast. Sprint running requires athletes to overcome inertia and accelerate from a stationary start to a high maximal velocity [2]. From a mechanical perspective, the ability to complete this movement task requires the athlete to apply a large amount of force and power in the horizontal direction at an increasing running velocity [3]. Although sprint mechanical characteristics have been assessed in various athletic populations in cross-sectional studies [4,5], there is a paucity of longitudinal research investigating individual mechanical changes in sprint athletes in response to specific periods of training. An analysis of sprint mechanical characteristics and performance is therefore of interest to practitioners as it may provide greater insight into training program design and periodization structure of sprint training and competition.

To quantify the mechanical determinants which underpin sprint performance, a field method known as force-velocity (F-v) profiling has been proposed by Samozino et al. [3]. Using an inverse dynamics approach to the body center of mass, the field method describes the mechanical output of over-ground maximal sprint running by modelling position-time data to indirectly estimate the underlying mechanical properties (i.e., forces) which produced the sprint performance [6]. The key mechanical variables obtained from sprint F-v profiles include theoretical maximal force (F_0_), theoretical maximal velocity (v_0_) and theoretical maximal power (P_MAX_) [3], which determine the intercepts of the inverse linear F-v relationship, and the parabolic relationship between power and velocity (P-v) [3].

The mechanical characteristics obtained by sprint force-velocity and power-velocity data can be used as a quantitative approach to improve the planning of sprint training to influence sprint outcomes during competition. The aim of sprint athletes who compete in traditional track events is to cover the competition distance (i.e., 100-meter) in the shortest time possible, however the aim of the coach is to periodize the training load and content to ensure the athlete produces their best performance at key times in the year, for example national championships. Furthermore, at different stages of the year, the training focus will likely change from attempting to improve various bio-motor abilities including strength and power, to more sprint-specific foci including acceleration, maximal velocity and speed endurance [7], a planning process known as periodization. Periodization of physical training has been identified as key to developing physiological and neuromuscular adaptations to maximize performance at specific periods during the training year [7]. Despite its recent widespread use in team sport to differentiate between ability level, field position and to individualize training strategies [5,8,9,10], an investigation into changes to mechanical characteristics in sprint athletes across a training year is yet to be explored.

Recent evidence has highlighted the importance of maximal power (P_MAX_) during the sprint action and the influence of individual F-v characteristics (i.e., S_FV_) to sprint acceleration performance [11]. Therefore, it would be useful information for sprint practitioners to understand mechanical changes across the training year and the relationships with sprint outcomes. Previous longitudinal case studies of junior (7 weeks, 100-meter personal best: 10.89 ± 0.21 s) and senior level (5 months, 100-meter personal best: 10.16 ± 0.16) sprinters focused on strength training and its effect on sprint performance [12], plus changes to step kinematics in response to periodized training [13]. Sprint performance changes in junior athletes were deemed inconclusive; however, it was hypothesized changes to performance in senior elite athletes was explained by the periodization of specific training components which was associated with an increase in force production, along with the ability to produce force rapidly leading to increases in step velocity and frequency during phases of low volume resistance training and high-intensity sprint training [13]. However, to the authors’ knowledge, no research exists examining changes to mechanical characteristics and the sprint F-v profile in national level sprint athletes across a training year.

Therefore, the aim of this case study was to investigate how sprint mechanical characteristics change across a track and field season (~45 weeks) in two male sprint athletes who qualified for their national championships. A secondary aim was to explore how periodized sprint training influences mechanical and spatio-temporal characteristics, step kinematics and sprint performance outcomes. We hypothesized that, as the periodization model changed between training phases and the mechanical load was reduced [7], it would likely result in improved sprint outcomes due to an enhanced F-v profile, plus optimized step kinematics for each athlete during 100-meter performance, however inter-athlete differences would be evident based on initial F-v characteristics and level of performance.

## 2. Materials and Methods

### 2.1. Participants

Two male sprint athletes who qualified for their national track and field championships (2021–22) in the 100-meter sprint event volunteered to participate in this study. Both athletes (Athlete 1: 22 years, 1.83 m, 81.1 kg, 100-meter time: 10.47 s; Athlete 2: 19 years, 1.82 m, 75.3 kg, 100-meter time: 10.81 s) met the inclusion criteria of completing a minimum of 10 sprint force-velocity assessments across the training and competition period. Further inclusion criteria included participants aged over 18 years of age. Exclusion criteria maintained that participants needed to be six-months free of musculoskeletal injuries which may prevent them from performing maximal effort sprints. The study was conducted in accordance with the Declaration of Helsinki, and the protocol was approved by the Social and Behavioral Research Ethics Committee at Flinders University (Ethics App Number: 8146). Personal best data and World Athletics points during the past 12 months of competition were collected from World Athletics [14] to establish a baseline for the performance levels of both athletes (100 m: 10.81 ± 0.42/895 ± 56.5 points, 200 m: 21.98 ± 1.01/898 ± 91.9 points).

### 2.2. Study Design

A case study design was used to monitor the sprint athletes from when they began their general preparation phase training at the end May 2021 and were followed through to the national championships at the start of April 2022 (~45-weeks). During this period, the athletes completed 12 (Athlete 1) and 11 (Athlete 2) force-velocity assessments, respectively, while also competing in 100-meter and 200-meter events (Table 1).

Training components including acceleration, speed, speed endurance and strength endurance, were periodized across the year to ensure the development and retention of specific physiological and neuromuscular adaptations [15,16]. The structure of training was defined by the two track and field coaching staff working with Athlete 1 and Athlete 2 and included running based sessions on grass fields, hills and synthetic tracks, plyometrics, along with gym-based resistance training sessions focused on developing aspects of the force-velocity continuum [17]. Typical training cycles and periodization of training components for the season are outlined in Table 2. During the preparation (PREP) phase, a 3:1 summated step loading model of periodization, Figure 1A, was implemented which allows for progressive overload of training modalities across three microcycles (~21 days), which is then followed by one microcycle (~7 days) of unloading, i.e., reduced training load [7,18,19]. The unloading period provides time for athlete regeneration and physiological adaptations to occur, while limiting the potential for overtraining [18]. Furthermore, the step-loading model of periodization also adds an aspect of inter-mesocycle contrast which may increase and stimulate adaptation(s) across the season [18]. The competition (COMP) phase was characterized with an undulating periodization model (also referred to as non-linear periodization), Figure 1B, across the mesocycle (~4 weeks) [20]. Undulating periodization provides more frequent changes to stimuli (i.e., volume, intensity) which have been reported to be more conducive to optimize gains in strength [20]. During the COMP phase, this approach to periodization has been implemented to provide a micro-dosing effect to training prior to reducing the training load ahead of a competition [21].

### 2.3. Methodology

Sprint F-v assessments occurred outdoors on synthetic running tracks during training sessions with Athlete 1 and Athlete 2 completing 12 and 11 assessments, respectively. No wind measurements were obtained. Body mass and environmental conditions (i.e., ambient temperature, barometric pressure) were collected on the day of each sprint F-v assessment due to its effect on F-v profile calculation. The biomechanical model to establish the F-v profile has previously been reported [3] and validated [22] when compared with direct measurement of ground reaction forces (GRF) from in-ground force plates and has been used in previous interventional studies [23]. Position-time data from the electronic timing games were used in a custom-made Microsoft Excel spreadsheet [24] to derive and model all force-velocity variables using the equations developed by Samozino et al. [3]. Recent explanations on the procedures used to determine sprint F-v characteristics are provided by Morin et al. [22].

Prior to the sprint F-v assessment, a standardized 45 min warm-up consisting of light jogging, dynamic running-based drills and movements, and 4–8 linear accelerations, over 10–40 m, progressing from sub-maximal to maximal was undertaken by each participant. Individually, participants then performed 30-meter maximal sprint efforts from either a four-point start or from starting blocks, wearing track spiked shoes. For each force-velocity assessment, the average splits times (i.e., 0–10 m, 0–20 m, 0–30 m) across three trials was used for reliability purposes and to determine the minimal detectable change in performance, in line with previous research [25,26]. Timing of sprint efforts were collected with electronic timing gates (Freelap Timing System, Fleurier–Switzerland). The Freelap Timing System is an electronic timing system which records the position-time data via a radio frequency connection between an antenna located in the FxChip on the athlete, and the transmitter on the track (Tx Junior Pro). The radio frequency transmission field is suggested to be 0.80 m by the manufacturer. Timing began when the athlete moved their hand off the touch pad resting on the ground (Tx Touch Pro), with split times recorded at each 10-meter interval once the athlete passed the timing gate (Tx Junior Pro Transmitter).

The FxChip was positioned on the athletes at the midline of the waistbelt, adjacent to the anterior superior iliac crest (ASIS). Specifications for setting up the touch pad and timing gates are detailed in Figure 2. The reported benefits of using a ‘touch-pad’ approach to start the timing system is a possible reduction in the body swing and momentum gathered prior to the sprint start which may occur in a standing start [27]. Previous research using a ‘touch pad’ reported strong between-test reliability, Intraclass Correlation Coefficient (ICC) = 0.92, and a typical error of 0.03 s over a 10-meter sprint distance, yet the authors noted the lack of familiarization of the starting technique with junior rugby players [27]. At the conclusion of each sprint effort, electronic timing gate data was sent via Bluetooth to an application (MyFreelap) on a smartphone device. Reaction time is not included in the total sprint time, which at world class level is typically 0.17 − 0.18 ± 0.03 s [28]. Timing gate data was also provided as feedback to athletes at the conclusion of each sprint effort. Between each sprint effort there was 5 min passive recovery period to ensure readiness before the next sprint and to limit fatigue.

The training year was periodized into two categories for statistical analysis: PREP (i.e., general and specific preparation phases—a focus on preparing the athletes for competition) and COMP (i.e., competitive phase—the focus is on achieving performance outcomes leading into state and national championships) [7,19]. The PREP phase was a 6-month period from June to December, while the COMP phase was a 3-month period from January to March. Split times were collected across the season (PREP and COMP) using timing gate data, along with body mass, standing stature and environmental conditions (i.e., barometric pressure, temperature), which were then imported into a custom-made Microsoft Excel spreadsheet [24] to determine the sprint mechanical parameters. Step kinematics were analyzed according to the methodology by Salo et al. [29] and independently verified by authors (DH and RVT) using video analysis software (Kinovea v0.9.5) [30] to determine average step length and step frequency across all 100-meter performances accessible on video across the season (Athlete 1, n = 6, Athlete 2, n = 8).

### 2.4. Statistical Analyses

Statistical analyses were determined from input into Microsoft Excel spreadsheets [31] plus coded in R (v3.6.1; R Foundation for Statistical Computing, R Core Team, Vienna, Austria), in the RStudio environment (v1.2.519; RStudio, Inc., Boston, MA, USA) using various statistical packages. All descriptive data are presented as mean ± standard deviation (SD) for force-velocity and spatio-temporal variables and were assessed for normality and variance using the Shapiro-Wilks and Levene’s test, respectively. Intraclass correlation coefficient (ICC) with 95% confidence limits, using a two-way random effect model (absolute agreement) and coefficient of variation (CV) were used to assess relative and absolute reliability of force-velocity, and spatio-temporal variables across the PREP phase only [32]. The thresholds for evaluation of intraclass correlation coefficients were quantified using the following scale: 0.20–0.49 low, 0.50–0.74 moderate, 0.75–0.89 high, 0.90–0.98 very high and ≥ 0.99 extremely high [33]. Previous biomechanical studies reported variables with a CV within the range of 10% as reliable [34], therefore acceptable reliability was determined with a coefficient of variation (CV) ≤ 10% [35] and ICC > 0.70 [36,37,38]. To account for typical fluctuations in sprint performance across each phase of training (PREP and COMP), the minimal detectable change (MDC), using 90% confidence intervals, was used to determine the minimum level of change necessary to represent a ‘true’ performance change, rather than random measurement error. MDC was calculated as 1.645 x Standard error of measurement (SEM) × √2 [39,40], from the average of sprint F-v profile variables collected during the PREP phase. The MDC% was defined as (MDC/$\overset{-}{x}$) × 100 [41]. Pearson’s product-moment correlation coefficient (Pearson’s *r*) was used to determine relationships between F-v variables and split times. The criteria to interpret the strength of the *r* coefficients were as follows: trivial (<0.1), small (0.1–0.3), moderate (0.3–0.5), high (0.5–0.7), very high (0.7–0.9), or practically perfect (>0.9) [33]. A one-way ANOVA with repeated measures was conducted to identify within-athlete changes between training phases. Within-athlete effect sizes (Cohen’s d) between training phases were determined with 95% confidence limits. Magnitudes of effect size changes were interpreted using the following values: trivial (<0.20), small (0.20 ≤ 0.60), moderate (0.60 ≤ 1.20), large (1.20 ≤ 2.00) and extremely large (>2.00) [42]. Linear regression analysis was also used to determine the relationship between 100-meter competition performance and step length (SL) and step frequency (SF). An alpha value of *p* ≤ 0.05 was used to indicate statistical significance.

## 3. Results

Shapiro-Wilks and Levene’s tests confirmed normality and homogeneity of variance for all F-v and spatio-temporal variables. Absolute and relative reliability, minimal detectable change (MDC) and standard error of measurement (SEM) data for force-velocity and spatio-temporal (split-times) variables for both athletes are presented in Table 3. Based on the F-v and spatio-temporal results from the PREP phase, intraclass correlation coefficients (ICC) and coefficient of variation (CV%) were almost all within acceptable limits (ICC: 0.73–0.98, CV%: 0.3–4.6) suggesting a high-level of reliability for both athletes when analyzing three sprint trials. The minimal detectable change (%) across F-v variables ranged from 1.3 to 10.0% while spatio-temporal variables were much lower, from 0.94 to 1.48%, with Athlete 1 showing a higher ‘true change’ in performance across the season compared to Athlete 2.

Descriptive data for force-velocity and spatio-temporal (split-times) variables for both athletes are presented as mean ± standard deviation (SD) in Table 4. Changes to F-v and P-v relationships between phases are highlighted in Figure 3. Athlete 1 showed significantly large within-athlete effects between phases for relative P_MAX_, v_0_, RF_MAX_, V_MAX_, and split time from 0 to 20 m and 0 to 30 m (−1.70 ≤ ES ≥ 1.92, *p* ≤ 0.05), which coincided with new personal best performances over both sprint distances during the COMP phase (100-meter: 10.47 s, 1050 pts) (Table 4, Figure 4(A1)). Athlete 2 reported significant differences with large effect for relative F_0_ only (ES: ≤ −1.32, *p* ≤ 0.01), which also led to new performance bests over 100-meter (10.81 s, 943 points) during the COMP phase (Table 4, Figure 4(A2)). Both athletes also reported statistically significant increases in maximum ratio of forces (RF_MAX_) (ES: ≤ −1.28, *p* ≤ 0.05). No significant changes to body mass were noted between phases (*p* ≥ 0.05).

During the PREP phase, both athletes showed high negative correlations with relative F_0_ and P_MAX_ and split time from 0 to 10 meters (r = −0.83, *p* ≤ 0.02), while during the COMP phase both athletes reported a higher correlation with v_0_ and 0 to 30 m which coincided with sprint performance outcomes during competition (Figure 5). Correlation and significance data between variables is available in Appendix A. The relationship between S_FV_, D_RF_, Tau and 0 to 30 m was also stronger during the COMP phase (Figure 5). An analysis of 100-meter performance and step kinematics highlights the reliance Athlete 1 (Figure 6(A1,A2)) has on step length to achieve faster sprint times (r = −0.95, *p* = 0.01), whereas Athlete 2 showed similar relationships between both step length (r = −0.72, *p* = 0.04) and step frequency (r = −0.70, *p* = 0.06) and 100-meter performance, however only step length achieved significance (Figure 6(B1,B2)). Non-significant changes were evident for S_FV_ and D_RF_ across the training year.

## 4. Discussion

The aim of this case study was to explore the mechanical changes to the sprint F-v profile and sprint outcomes across a track and field season in two 100-meter athletes who qualified for the national championships. To the authors knowledge, this is the first study to use longitudinal training data to investigate the relationship between F-v variables and sprint performance outcomes across a 10-month period. We believe the information presented including typical training microcycles, force-velocity and spatio-temporal variables, along with step kinematics, provide a holistic and transparent view of the changes which occur in response to periodized sprint training.

Our key findings are as follows: (a), when comparing the PREP and COMP phases, Athlete 1 showed an enhanced F-v profile due to significant changes to relative P_MAX_, v_0_ and improved F_0_, whereas Athlete 2 reported significant changes to F_0_ and improved P_MAX_ thereby demonstrating a more ‘force-oriented’ F-v profile, (b) positive mechanical changes and improved sprint performance observed during the early COMP phase was significantly correlated with increased step length and favorable step frequency, and (c) inter-athlete differences were observed for correlations between F_0_ and P_MAX_ and 0–10 m in the PREP phase, and v_0_ and 0–30 m during COMP phase.

In reference to our hypothesis, the longitudinal nature of this study primarily identifies the influence specific sprint training stimuli and periodization models have on sprint F-v characteristics, thereby highlighting the F-v profile adheres to the SAID principle (Specific Adaptations to Imposed Demands) [43]. Once the periodization model changed between the PREP and COMP phase, sprint mechanical characteristics were enhanced in both athletes. This confirmed our hypothesis. With respect to the F-v profile with the highest force value for each athlete, relative F_0_ (8.13–8.92 N.kg^−1^), V_MAX_ (9.67–10.49 m.s^−1^) and P_MAX_ (21.11–24.78 W.kg^−1^) were maximized during the COMP phase within a 35-day period between January and March with changes evident in F-v profiles between phases. For Athlete 1, when relative P_MAX_ increased during the COMP phase it resulted in a season’s best 100-meter performance (10.47 s), whereas Athlete 2 had similar performance outcomes (10.84 s) in response to an increase in relative F_0_ (Figure 4B1,B2). Samozino et al. [11] have recently showed sprint acceleration performance, irrespective of distance, is directly related to the average external power output produced over the entire targeted distance, therefore from a mechanical perspective, the 100-meter performance differences, and changes in pre-post F-v profiles between athletes may be expected due to Athlete 1 demonstrating superior P_MAX_, and significant changes to v_0_ in the COMP phase. Furthermore, previous studies focusing on longer sprint accelerations (i.e., 40–100-meter) identified both P_MAX_ and v_0_ as key determinants of performance [44,45,46,47].

Significant mechanical changes also appear to coincide with a change in periodization models. A step-loading periodization model in the PREP phase had a focus on speed endurance (i.e., high intensity efforts for 7–15 s in duration), strength endurance (i.e., hill work, moderate to high intensity efforts for 15–45 s in duration) and a greater number of strength and conditioning sessions, whereas during the COMP phase an undulating periodization model placed a greater focus on acceleration and speed work (i.e., maximum intensity and velocity efforts ≤7 s in duration), plyometrics, less strength and conditioning sessions, with an overall higher intensity and lower volume (meters) (Table 2). When comparing both athletes, during the transition period from PREP to the COMP phase, although greater for Athlete 1, it could be surmised the upward trend in P_MAX_ reflects a reduction in training density, less mechanical load, greater recovery time and an emphasis on neuromuscular development via velocity specific training modalities (Figure 4B1). This change in periodization model from training quantity (i.e., volume) to training quality (i.e., speed-specific intensity), although relatively typical during sprint training programs [43], appears to have been also led to personal best performances during 100-meter competitions. 

Both athletes in this study showed a significant relationship between step length and 100-meter performance (r ≥ −0.72, *p* ≤ 0.01), highlighting their reliance on this component to achieve faster velocities, however Athlete 2 did also demonstrate a moderate non-significant correlation with step frequency (r ≥ −0.70). Associations between step length (2.46–2.60 m) and sprint performance have previously been reported in elite level male sprinters (10.18–10.52 s), highlighting key differences in finishing position based on step length [48]. Other research has acknowledged a significant relationship between step length and sprint velocity (r = 0.73), and a negative interaction effect between step length and step frequency (r = −0.78) based on individual biomechanical and kinematic characteristics [48,49]. Contradictions to these findings have also been presented [13] identifying a clear association between step frequency (group mean: 4.85 Hz) and 100-meter performance (10.16 ± 0.16 s), with lower step frequency noted in specific training blocks (4.34 Hz). It has previously been suggested that step length is more related to increased force production, whereas step frequency is associated with higher rates of force production during ground contact and leg turnover requiring greater neural adaptations [29,50], which may also be a reflection of training load and training content during the COMP phase. It could therefore be concluded, that limiting the volume of speed endurance and strength endurance leading into important competitions has maximized mechanical characteristics and step kinematics necessary to drive 100-meter performance outcomes. Moreover, when attempting to plan training for the successive training year, placing a greater emphasis on acceleration and speed work during these periods at the expense of other training modalities may enhance P_MAX,_ as these training modalities would encourage higher V_MAX_ and therefore potentially further optimize step kinematics and the F-v profile and provide greater improvements in sprint performance. Despite differences in previous studies regarding step kinematics, this may be accounted for due to subject population and performance level of the athlete (i.e., faster athletes).

Correlations between F-v and spatio-temporal variables across the training year identify how the training phase affects F-v characteristics of each athlete differently. Both athletes demonstrated similar correlations between F_0_ and P_MAX_ from PREP to COMP phase however stronger correlations between spatio-temporal variables and v_0_ exist once the periodization structure moved into the COMP phase (Figure 5). This is likely a result of the change in training focus, but more importantly the frequent demand for maximal velocity efforts during competitions. The decrement in ratio of forces (D_RF_) or mechanical effectiveness [3] of both athletes also showed stronger correlations in the COMP phase compared to the PREP phase, potentially due to neuromuscular adaptation and the ability to continue producing a high level of horizontally directed force across the sprint effort at higher running velocities. Adaptations for D_RF_ have been observed in sprint athletes with similar 100-meter performance levels of those in this case study [51].

It is interesting to note, for both athletes, a downward trend in body mass (Athlete 1: −2.6%, Athlete 2: −1.9%) from the beginning of the PREP phase until the early COMP phase also coincided with positive mechanical changes and performance outcomes (Appendix A). Body mass is a key consideration for sprint performance due to fundamental Newtonian laws of motion and the energy cost of accelerating a higher mass. Uth [52] has previously identified elite male sprinters having body mass values of 77 ± 7 kg, however it is the change and improvement in relative mechanical values and the ability to apply mass specific force (i.e., force and power per kilogram of body mass) which is of greater importance during maximal velocity sprinting [53].

A novel aspect of this case study is to explore the variability and minimal detectable change (MDC) in respect to sprint force-velocity variables across the training year. Based on the average of F-v variables across the PREP phase, Athlete 1 and Athlete 2 exceeded the MDC in 82% and 55% of sprint force-velocity and spatio-temporal variables, respectively, suggesting a true change in performance occurred beyond the measurement error (Table 3). Previous research using MDC to detect changes in F-v characteristics and sprint performance in junior Australian football players suggests this is an appropriate measure to determine improvements are a result of the training interventions rather than error [25]. The MDC for the same variables is much lower in magnitude in this case study compared to previous research, however this is likely accounted for in difference in sprint performance between the two population groups. 

Interestingly, Athlete 1 tested positive to COVID-19 on 18/FEB/2022, therefore beginning a 10-day isolation period in his home, as per local government regulations. During this time, the athlete was quite ill and only limited training could be done including basic bodyweight resistance training and stationary bike intervals. Upon resuming training, an obvious level of fatigue was evident resulting in slower running times. This appears to be reflected in a decline in relative F_0_ (−9.51%), v_0_ (−0.06%) and P_MAX_ (−9.22%) between the F-v profiles collected before and after the illness (Figure 4B1), along with recording the slowest 100-meter performance of their season, 10.66 (19/MAR/22)(Figure 4A1). Analysis of step kinematics identifies a reduction in step length during this performance period, which is likely a result of a reduction in force production while sprinting (Figure 4C1). Commentary on the impacts of COVID-19 and sport performance has centered on physical and mental health, with authors suggesting the reduced training frequency, potential loss in muscle function and emotional health from isolation to have a negative impact on performance outcomes once returning to training and competition [54,55,56].

Due to the exploratory nature of this case study, the authors’ identified several limitations. Firstly, the small sample size of athletes (n = 2) provides a narrow cross-section of sprint F-v and performance data from which to analyze. Post-hoc analysis using the following test details: ANOVA: repeated measures, within factors, with an effect size of 0.5, alpha of 0.05, provides a power level of only 0.29, which highlights differences between the means will only be detected 29% of the time. To achieve 0.8 power, we would require six participants in this study. This may limit the conclusions outlined below as the case study is underpowered. Secondly, the part-time status of the athletes and the availability of training hours on synthetic tracks made it necessary to conduct F-v assessments at different hours of the day (i.e., morning and late evening) across the training year, reflecting the dynamic considerations of the practitioners. Additionally, despite several force-velocity assessments occurring as part of a designated testing session, most assessments were collected as part of a typical training session within the mesocycle. Thirdly, recent research [57] has suggested a time correction (+0.21) is necessary for calculating accurate F-v profiles when comparing electronic timing gate data with more precise technology such as an optical laser gun. Despite the difference in methodology and data collection in this study, this should be taken into consideration. Finally, future research should investigate sprint athletes involved in national finals or international competition to monitor the change in mechanical, spatio-temporal and sprint kinematic variables leading into a major competition. 

## 5. Conclusions

This is the first longitudinal study to investigate how a periodized sprint training program influenced force-velocity characteristics, step kinematics and 100 m sprint performance in national level sprint athletes. For both athletes, once the periodization model changed between training phases sprint mechanical characteristics were enhanced and increases in step length showed greater correlations with 100 m sprint performance. The findings of this study may provide practitioners with greater insight into training program design and periodization structure for athletes of similar performance levels, plus identify the underpinning mechanical characteristics and step kinematics affecting sprint outcomes leading into national championships. Practitioners may also use the results of this study to anticipate changes to sprint performance at different phases of the training year, while also identifying which periodization models and sprint mechanical characteristics lead to improved performance outcomes for their athletes.

## Figures and Tables

**Figure 1 ijerph-19-15404-f001:**
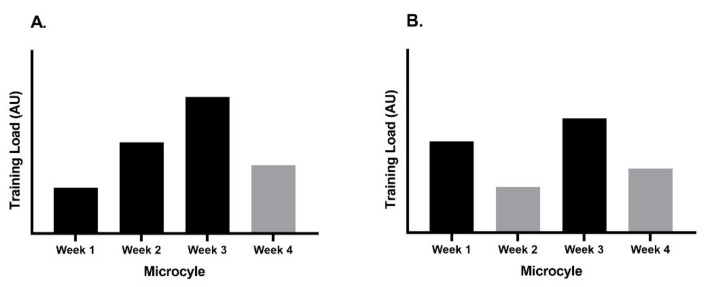
Periodization models used across the training year. (**A**): represents the summated step-loading periodization model for the preparation phase; (**B**): represents the undulating periodization model during the competition phase.

**Figure 2 ijerph-19-15404-f002:**
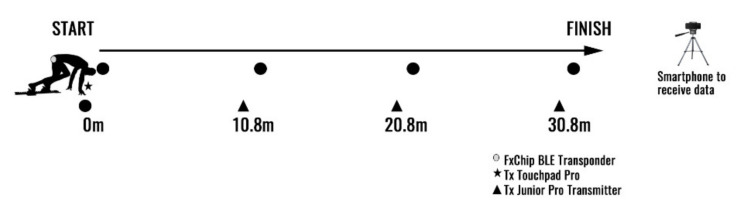
Electronic timing gate (Freelap) setup to record split times (10-meter intervals) from 0–30 m.

**Figure 3 ijerph-19-15404-f003:**
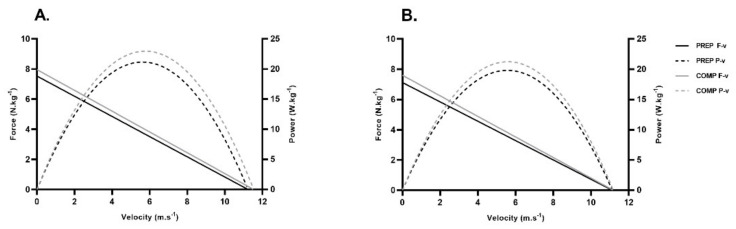
Sprint force−velocity (F−v) and power−velocity (P−v) relationships between the PREP and COMP phase. (**A**): Athlete 1, (**B**): Athlete 2.

**Figure 4 ijerph-19-15404-f004:**
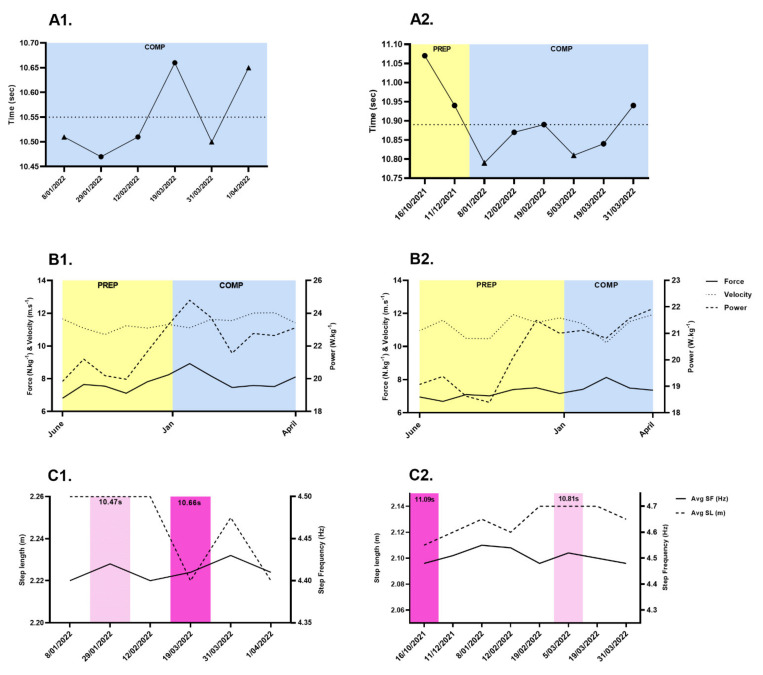
Sprint performance, F−v variables, and step kinematics across the training year. (**A1**): Athlete 1 100−meter performances, (**A2**): Athlete 2 100−meter performances (PREP = preparation phase, COMP = competition phase. Dotted line: average of performances. Circle: legal performance, triangle: wind−aided performance (>+2.0 m.s^−1^)); (**B1**): Athlete 1 force, velocity, and power changes across the training year, (**B2**): Athlete 2 force, velocity, and power changes across the training year; (**C1**): Athlete 1 step kinematics during 100−meter competitions; (**C2**): Athlete 2 step kinematics during 100−meter competitions. (Dark shade column = slowest performance of season, light shade column = season’s best).

**Figure 5 ijerph-19-15404-f005:**
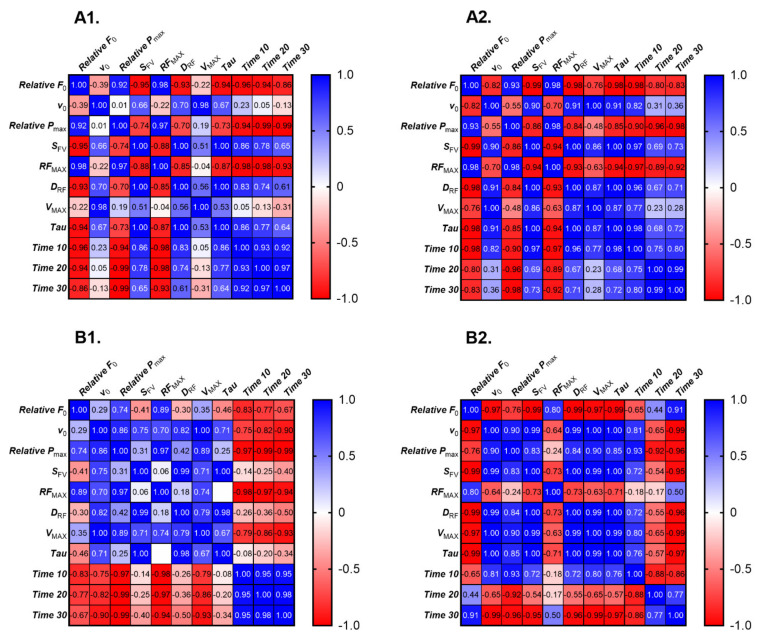
Correlation matrix between F−v variables and spatio−temporal variables. (**A1**): Athlete 1 PREP, (**A2**): Athlete 1 COMP; (**B1**): Athlete 2 PREP, (**B2**): Athlete 2 COMP.

**Figure 6 ijerph-19-15404-f006:**
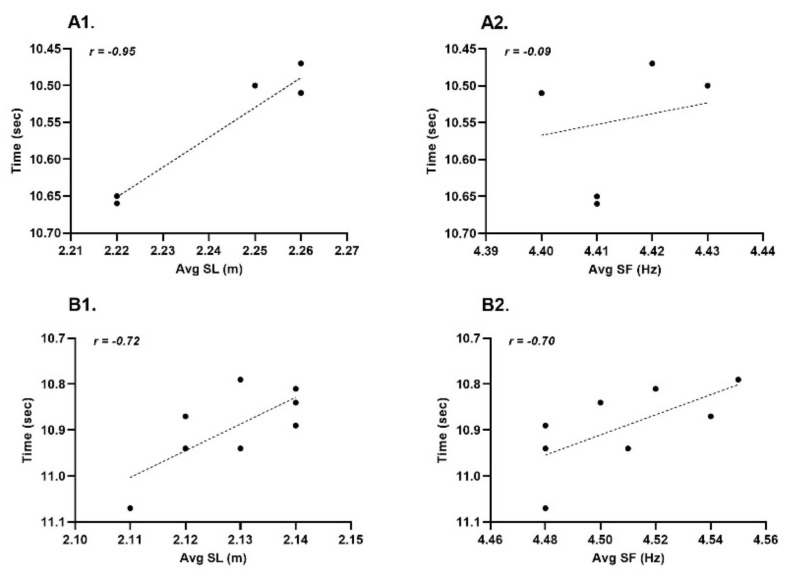
Individual 100−meter competition times as a function of step length (SL) and step frequency (SF). Athlete 1: (**A1**,**A2**); Athlete 2: (**B1**,**B2**). Note that the y−axes have been inverted because faster times highlight improved performance. Due to inverted y−axes, the direction of *r* values does not match the visual impression.

**Table 1 ijerph-19-15404-t001:** Timeline and number of force-velocity assessments and competitions across the training year.

Date	Phase	Type	Athlete 1	Athlete 2
June-21	PREP	FV	1	1
July-21	PREP	FV	2	2
August-21	PREP	FV	2	2
October-21	PREP	100 m/200 m	-	3
November-21	PREP	FV	1	1
November-21	PREP	100 m/200 m	-	1
December-21	PREP	100 m/200 m	-	1
December-21	PREP	FV	1	1
January-22	COMP	FV	1	1
January-22	COMP	100 m/200 m	4	3
February-22	COMP	FV	1	1
February-22	COMP	100 m/200 m	2	4
March-22	COMP	FV	2	2
March-22	COMP	100 m/200 m	2	3
April-22	COMP	FV	1	-
April-22	COMP	100 m/200 m	2	2

PREP = preparation phase, COMP = competition phase, FV = force-velocity profile, 100 m/200 m = competition performance.

**Table 2 ijerph-19-15404-t002:** Typical training microcycles across preparation phases during the training year.

Preparation Phase (General: June–September)							
DAY	SUNDAY	MONDAY	TUESDAY	WEDNESDAY	THURSDAY	FRIDAY	SATURDAY
INTENSITY	MODERATE	MODERATE	MODERATE	MODERATE-HARD	MODERATE	EASY	MODERATE-HARD
LOCATION	GRASS INCLINE	GRASS FIELD	WEIGHTROOM	TRACK	WEIGHTROOM	POOL/BEACH	TRACK
MAIN SESSION	AM Hill runs	PM Speed Endurance	PM Accumulation-Strength-Speed (UB)	PM Special Endurance	PM Accumulation-Speed-Strength (LB)	Regeneration	AM Acceleration/Speed Weightroom (TB) Maximal effort
Preparation Phase (Specific: October–December)							
DAY	SUNDAY	MONDAY	TUESDAY	WEDNESDAY	THURSDAY	FRIDAY	SATURDAY
INTENSITY	MODERATE	EASY-MODERATE	MODERATE-HARD	MODERATE	HARD	EASY	MODERATE-HARD
LOCATION	WEIGHTROOM	GRASS FIELD	TRACK	WEIGHTROOM	TRACK	POOL/BEACH	TRACK
MAIN SESSION	AM Intensification -Strength-Speed (LB)	PM Varied-paced runs	PM Acceleration/ Special Endurance	PM Intensification-Speed-Strength (UB)	PM Maximal Velocity + Tempo	Regeneration	AM Acceleration/ Speed Endurance
Competitive Phase (January–March)							
DAY	SUNDAY	MONDAY	TUESDAY	WEDNESDAY	THURSDAY	FRIDAY	SATURDAY
INTENSITY	EASY	EASY-MODERATE	MODERATE-HARD	MODERATE	MODERATE	EASY	MODERATE
LOCATION	WEIGHTROOM	GRASS FIELD	TRACK	WEIGHTROOM	TRACK	POOL/BEACH	TRACK
MAIN SESSION	PM Strength Circuits (TB)	PM Varied-paced runs	PM Acceleration/Speed	PM Power (TB)	PM Maximal velocity + Tempo	Regeneration	PM Competition

(UB = Upper body, LB = Lower body, TB = Total body).

**Table 3 ijerph-19-15404-t003:** Reliability measures and minimal detectable change for force-velocity and spatio-temporal variables across the training year.

Variable	Relative F_0_ (N.kg^−1^)	v_0_ (m.s^−1^)	Relative P_MAX_ (W.kg^−1^)	Relative S_FV_ (N.s.m^−1^.kg^−1^)	RF_MAX_ (%)	D_RF_ (%.m.s^−1^)	V_MAX_ (m.s^−1^)	Tau	Split Time 0–10 m (s)	Split Time 0–20 m (s)	Split Time 0–30 m (s)
Athlete 1											
ICC	0.94 (0.89, 0.96)	0.73 (0.51, 0.88)	0.94 (0.85, 0.98)	0.87 (0.73, 0.95)	0.96 (0.91, 0.98)	0.85 (0.70, 0.94)	0.82 (0.62, 0.94)	0.87 (0.73,0.95)	0.89 (0.77, 0.96)	0.98 (0.94 0.99)	0.91 (0.81, 0.97)
CV (%)	1.83	1.69	0.99	3.36	0.55	3.44	1.40	3.06	0.57	0.31	0.30
SEM	0.11	0.18	0.31	0.01	0.002	0.002	0.12	0.04	0.01	0.01	0.02
MDC	0.32	0.51	0.86	0.05	0.008	0.005	0.32	0.10	0.03	0.03	0.06
MDC%	4.24	4.56	4.08	7.46	1.66	8.33	3.06	7.35	1.48	0.95	1.43
Athlete 2											
ICC	0.89 (0.76, 0.96)	0.86 (0.70, 0.95)	0.96 (0.87, 0.98)	0.80 (0.26, 0.94)	0.96 (0.91, 0.98)	0.82 (0.36, 0.94)	0.88 (0.72, 0.96)	0.81 (0.61, 0.93)	0.93 (0.81, 0.98	0.97 (0.95, 0.98)	0.97 (0.95,0.98)
CV (%)	2.31	2.23	0.68	4.50	0.64	4.61	1.88	3.94	0.49	0.30	0.28
SEM	0.09	0.22	0.28	0.02	0.003	0.002	0.17	0.03	0.01	0.01	0.01
MDC	0.26	0.65	0.79	0.06	0.006	0.006	0.48	0.09	0.03	0.03	0.05
MDC%	3.65	5.78	3.95	9.37	1.30	10.00	4.61	6.29	1.44	0.94	1.17

ICC = intraclass correlation coefficient; CV = coefficient of variation; MDC = minimal detectable change, ICC are expressed with 95% confidence intervals.

**Table 4 ijerph-19-15404-t004:** Descriptive statistics for force-velocity and spatio-temporal variables across the training year.

Variable	Participant	PREP Mean ± SD	COMP Mean ± SD	Mean Difference, %Δ	Within-Athlete ES (+ 95% CL) (PRE-COMP)	*p* Value
**Relative F_0_ (N.kg^−1^)**	Athlete 1 Athlete 2	7.53 ± 0.50 7.12 ± 0.27	7.96 ± 0.56 7.60 ± 0.35	0.43, 5.77 0.48, 6.33	−0.81 (−2.55, 0.92) −1.56 (−3.17, 0.03)	0.19 0.03 *
**v_0_ (m.s^−1^)**	Athlete 1 Athlete 2	11.18 ± 0.31 11.23 ± 0.59	11.62 ± 0.35 11.27 ± 0.72	0.44, 3.81 0.04, 0.29	−1.32 (−3.44. 0.79) −0.05 (−1.46, 1.36)	0.04 * 0.94
**Relative P_MAX_ (W.kg^−1^)**	Athlete 1 Athlete 2	21.03 ± 1.32 20.00 ± 1.48	23.10 ± 1.09 21.36 ± 0.49	2.07, 8.99 1.36, 6.34	−1.70 (−3.79. 0.37) −1.08 (−2.59, 0.42)	0.01 ** 0.12
**Relative S_FV_** **(N.s.m^−1^.kg^−1^)**	Athlete 1 Athlete 2	−0.67 ± 0.05 −0.64 ± 0.03	−0.69 ± 0.06 −0.68 ± 0.07	−0.02, 1.80 −0.04, 6.42	0.20 (−2.40, 1.80) 0.80 (−0.66, 2.27)	0.73 0.23
**RF_MAX_** **(Maximum ratio of forces)**	Athlete 1 Athlete 2	0.48 ± 0.01 0.46 ± 0.01	0.49 ± 0.01 0.48 ± 0.002	0.01, 3.71 0.02, 3.24	−1.28 (−3.21, 0.63) −1.46 (−3.04, 0.11)	0.05 * 0.04 *
**D_RF_** **(Decrement in ratio of forces)**	Athlete 1 Athlete 2	−0.060 ± 0.00 −0.057 ± 0.00	−0.061 ± 0.00 −0.061 ± 0.01	0.001, 0.88 0.003, 5.97	0.10 (−1.50, 1.70) 0.70 (−0.75, 2.16)	0.87 0.29
**V_MAX_** **(Maximal horizontal** **velocity)**	Athlete 1 Athlete 2	10.43 ± 0.24 10.41 ± 0.49	10.84 ± 0.26 10.48 ± 0.57	0.41, 3.83 0.07, 5.69	−1.63 (−3.93, 0.65) −0.13 (−1.55, 1.28)	0.01 ** 0.84
**Tau** **(Relative acceleration)**	Athlete 1 Athlete 2	1.36 ± 0.10 1.43 ± 0.07	1.34 ± 0.11 1.36 ± 0.12	−0.02, 1.64 −0.07, 2.20	0.20 (−1.38, 1.78) 0.81 (−1.55, 2.28)	0.74 0.22
**Split time 0–10 m (s)**	Athlete 1 Athlete 2	2.02 ± 0.04 2.07 ± 0.04	1.96 ± 0.04 2.02 ± 0.01	−0.06, 2.72 −0.05, 2.15	1.20 (−0.61, 3.01) 1.10 (−0.40, 2.62)	0.06 0.11
**Split time 0–20 m (s)**	Athlete 1 Athlete 2	3.14 ± 0.07 3.19 ± 0.07	3.04 ± 0.05 3.11 ± 0.03	−0.10, 3.38 −0.08, 2.45	1.57 (−0.55, 3.70) 1.23 (−0.30, 2.76)	0.02 * 0.08
**Split time 0–30 m (s)**	Athlete 1 Athlete 2	4.18 ± 0.07 4.25 ± 0.11	4.05 ± 0.05 4.17 ± 0.06	−0.13, 3.11 −0.07, 2.03	1.92 (−0.18, 4.03) 0.83 (−0.63, 2.30)	0.007 * 0.22

PREP = preparation phase (general and specific), COMP = competitive phase. ES = effect size, CL = confidence limits. * = *p* < 0.05. ** = *p* < 0.005.

## Data Availability

Raw data and supplemental files for this article can be found online at DOI: 10.17605/OSF.IO/QCDKH.

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
