# Peer review of "Exploratory Analysis of Sprint Force-Velocity Characteristics, Kinematics and Performance across a Periodized Training Year: A Case Study of Two National Level Sprint Athletes"

_ijerph, 2022, doi:10.3390/ijerph192215404_

Round 1

Reviewer 1 Report

Exploratory analysis of sprint force-velocity characteristics, kinematics and performance across a periodized training year: A case study of two national level sprint athletes

I welcome studies that introduce novelty and applicability the important effects of explore changes to sprint force-velocity characteristics 10 across a periodized training year (45-weeks) and the influence on sprint kinematics and perfor- 11 mance in national level 100-meter athletes. In fact, I am some sympathy with the author's intentions. In addition, the authors provide a decent description of his potential benefits in soccer player with groin pain and the topic represents contemporary interest, and the scope of the work is appropriate for International Journal of Environmental Research and Public Health

From my point of view, the main strength of the manuscript are the results. The authors have followed properly analysis.  Apart from this general commentary about the manuscript, more details of some parts of the manuscript (strengths, weakness, and questions) are found hereafter. However, I would like the individual values also and not only the means.

Introduction

It is well written and structured. However, this paper needs more background. It is a good starting point to place the reader; However, it would be helpful to elucidate in the first paragraph of introduction why you investigate about this topic? In addition, I have not clear if all the actual literature was added? it would be interesting if the references was updated.

Methodology

The methodology clearly explained and justified. Can you add in the participants section a paragraph with the sample size calculated with G-power? I know that is a case study, but maybe the researchers need know how many people need to growth in this topic

-          Results

As it has been mentioned before, this section is the strongest part of the manuscript. The results are clear. In addition, I think that the results will be explained in the discussion also.

-          I know that R created the figures in colour. Maybe all the figure of Figure 4 and all the figures of Figure 5, you can put in colour. Because I understanding that is a colour map and add in each cells the r and the p.

Discussion and conclusion

Different aspect of conclusion should appear in the discussion section. The theoretical and practical implications of the research are vaguely mentioned at the end of paper.   I also suggest you are more specific in defining the rationale for your study. How will your findings help coaches, for example?
More soccer-specific in depth discussion and practical relevance of the results should be emphasized. If the paper will be allowed to be revised, I am happy to revise the discussion again thoroughly and provide some in-depth comments and suggestions

Author Response

Response to Reviewer 1 Comments

Thank you to all reviewers of this article. Not all suggested revisions were amended. All suggested revisions were fair yet we provided explanations of why we chose not to amend these. Those we did amend provided greater scope to the topic that ultimately benefits the reader.

Point 1:

Introduction

It is well written and structured. However, this paper needs more background. It is a good starting point to place the reader; However, it would be helpful to elucidate in the first paragraph of introduction why you investigate about this topic? In addition, I have not clear if all the actual literature was added? it would be interesting if the references was updated.

Response 1: Thankyou for this comment. We have revised the first paragraph and updated the references.  

Point 2:

Methodology

The methodology clearly explained and justified. Can you add in the participants section a paragraph with the sample size calculated with G-power? I know that is a case study, but maybe the researchers need know how many people need to growth in this topic

Response 2: Thankyou for this comment. We have added a section in the limitations section (line 473) regarding sample size and believe this is a more appropriate section to add this content We also believe there is a level of presumption the case-study will be under-powered (n=2).

Point 3:

Results

As it has been mentioned before, this section is the strongest part of the manuscript. The results are clear. In addition, I think that the results will be explained in the discussion also.

I know that R created the figures in colour. Maybe all the figure of Figure 4 and all the figures of Figure 5, you can put in colour. Because I understanding that is a colour map and add in each cells the r and the p.

Response 3: We have provided Figure 4 & 5 in color. We have provided a supplemental data file of p values for the correlation matrices. This is available at the url for data availability statement.

Point 4:

Different aspect of conclusion should appear in the discussion section. The theoretical and practical implications of the research are vaguely mentioned at the end of paper.   I also suggest you are more specific in defining the rationale for your study. How will your findings help coaches, for example?
More soccer-specific in depth discussion and practical relevance of the results should be emphasized. If the paper will be allowed to be revised, I am happy to revise the discussion again thoroughly and provide some in-depth comments and suggestions

Response 4: Thankyou for this comment. We have revised the conclusion to provide better context for practitioners.

Reviewer 2 Report

Sprining review

The authors state that the aim of this study is

to investigate how sprint mechanical characteristics change across a track and field season (~45 weeks) in two male sprint athletes who qualified for their national championships. A secondary aim was to explore how periodized sprint training influences mechanical and spatio-temporal characteristics, step kinematics and sprint performance outcomes.

This information is of interest to coaches and athletes in track and field as well as in a number of other sports in which speed is a performance determining variable. The design of this research study is a case study which would typically give us a much better view of how an individual adapts to training loads throughout a season while also giving us a lower number of subjects and less generalizability across a given population. The current study seems to have required a large time commitment and the researchers should be commended for their efforts.

Overall this study is well written and well designed. This topic is of interest to the practitioners as well as to the scientific community. Although case studies are typically difficult to get published I believe this is an important first step in this line of research. Below I have made some suggests based on figure and table legends and some basic grammar.

Intro

Line 52 - They key mechanical variables

Suggest changing to: The key mechanical variables.

Table 1

the text under table 1 states FVP = force-velocity profile but nowhere in the table is FVP present. I think  It is referring to FV

Figure 1

The text of figure 1 states ‘Figure 1. Periodization models used across the training year’ and shows two bar graphs labeled ‘A’ and ‘B’. Figures should be ‘stand alone’ so please state what bar graphs ‘A’ and ‘B’ represent. ie figure 1a represents the periodization model for the preparation phase and figure 1b represents the periodization model during competition.

Author Response

Response to Reviewer 2 Comments

Thank you to all reviewers of this article. All suggested revisions were fair and provided greater scope to the topic that ultimately benefits the reader.

Point 1:

Intro

Line 52 - They key mechanical variables

Suggest changing to: The key mechanical variables

Response 1: We have amended this sentence

Point 2:

Table 1

the text under table 1 states FVP = force-velocity profile but nowhere in the table is FVP present. I think  It is referring to FV

Response 2: We have amended the text under Table 1

Point 3:

Figure 1

The text of figure 1 states ‘Figure 1. Periodization models used across the training year’ and shows two bar graphs labeled ‘A’ and ‘B’. Figures should be ‘stand alone’ so please state what bar graphs ‘A’ and ‘B’ represent. ie figure 1a represents the periodization model for the preparation phase and figure 1b represents the periodization model during competition.

Response 3: We have amended the text under Figure 1
